# Sedation AND Weaning In Children (SANDWICH): protocol for a cluster randomised stepped wedge trial

Bronagh Blackwood ![ORCID],[1] Ashley Agus ![ORCID],[2] Roisin Boyle,[2] Mike Clarke ![ORCID],[3] Karla Hemming,[4] Joanne Jordan,[1] Duncan Macrae,[5] Daniel Francis McAuley ![ORCID],[1] Clíona McDowell,[2] Lisa McIlmurray,[1] Kevin P Morris,[6] Margaret Murray,[2] Roger Parslow,[7] Mark J Peters,[8,9] Lyvonne N Tume,[10] Tim Walsh,[11] On behalf of the Paediatric Intensive Care Society Study Group (PICS-SG)

For numbered affiliations see end of article.

**Correspondence to**
Professor Bronagh Blackwood;
b.blackwood@qub.ac.uk

Dr Margaret Murray;
MargaretX.Murray@nictu.hscni.net

## ABSTRACT

**Introduction** Weaning from ventilation is a complex process involving several stages that include recognition of patient readiness to begin the weaning process, steps to reduce ventilation while optimising sedation in order not to induce distress and removing the endotracheal tube. Delay at any stage can prolong the duration of mechanical ventilation. We developed a multicomponent intervention targeted at helping clinicians to safely expedite this process and minimise the harms associated with unnecessary mechanical ventilation.

**Methods and analysis** This is a 20-month cluster randomised stepped wedge clinical and cost-effectiveness trial with an internal pilot and a process evaluation. It is being conducted in 18 paediatric intensive care units in the UK to evaluate a protocol-based intervention for reducing the duration of invasive mechanical ventilation. Following an initial 8-week baseline data collection period in all sites, one site will be randomly chosen to transition to the intervention every 4 weeks and will start an 8-week training period after which it will continue the intervention for the remaining duration of the study. We aim to recruit approximately 10 000 patients. The primary analysis will compare data from before the training (control) with that from after the training (intervention) in each site. Full details of the analyses will be in the statistical analysis plan.

**Ethics and dissemination** This protocol was reviewed and approved by NRES Committee East Midlands—Nottingham 1 Research Ethics Committee (reference: 17/EM/0301). All sites started patient recruitment on 5 February 2018 before randomisation in April 2018. Results will be disseminated in 2020. The results will be presented at national and international conferences and published in peer-reviewed medical journals.

**Trial registration number** ISRCTN16998143.

## INTRODUCTION

On average more than 20 000 children are admitted to paediatric intensive care units (PICUs) in the UK and 65% of admissions to PICU require invasive mechanical ventilation (IMV) for acute respiratory failure.[1] Weaning and extubation from IMV are key steps in

### Strengths and limitations of this study

► Sedation AND Weaning In CHildren (SANDWICH) is the first large multicentre pragmatic randomised trial (approximately 10 000 children) evaluating a collaborative sedation and weaning protocol aimed at reducing the duration of invasive mechanical ventilation in critically ill children.
► From inception, SANDWICH has had strong involvement from medical and nursing staff, parents and patients, and a children's research advisory group.
► The trial has an embedded cost-effectiveness and process evaluation.
► The primary outcome is patient relevant and was proposed by parents and children during feasibility work.
► A limitation may be the practicality of achieving all signed research and governance approvals to enable sites to start at the same time within the required start-up time frame.

the child's recovery and indicate progression towards PICU discharge. Deferments in weaning impact on patient morbidity prolong PICU stay and bed availability.

Currently, there is no consensus on the optimal weaning approach from IMV in PICUs. Our feasibility study highlighted considerable variation in ventilator weaning practice: usually a slow reduction in ventilator support to a very low level prior to extubation and no test of early readiness for extubation on higher levels of support using a trial of spontaneous breathing.[2] Furthermore, nurses' roles are not optimally utilised to adjust ventilator settings due to lack of protocols to guide ventilator weaning and discontinuation.[3] In many PICUs, very few nurses are engaged in weaning, most PICUs suspend changes to ventilator settings overnight and weaning only happens during the day.[2]

Weaning from ventilation involves: (1) recognition that the child is ready to begin the weaning process, (2) steps to reduce ventilation while optimising sedation in order not to induce distress and (3) removing the endotracheal tube. Delay at any stage can prolong the duration of IMV; therefore, an intervention targeted at helping clinicians to expedite this process safely should reduce the harms associated with IMV. However, the judgement and experience of clinicians are critical in guiding weaning from ventilation, as our feasibility study showed, there is wide variation in sedation and ventilator weaning practices, junior staff are rarely involved in the process and use of weaning protocols is rare.[2]

A Cochrane review of weaning protocols in mechanically ventilated children highlighted only three randomised trials.[4] A two-centre trial (n=260), using an intervention incorporating daily screening and a spontaneous breathing trial (SBT), demonstrated a significant reduction of 32 hours (95% CI: 8 to 56 hours) in duration of IMV without additional harms.[5] The smaller pilot studies using computer-driven protocols showed non-significant effects on duration of IMV, but significant reductions in weaning times (106 hours, 95% CI: 28 to 184; and 21 hours, 95% CI: 9 to 32).[6 7] A recent paediatric multicentre cluster randomised trial in the USA (n=31 sites) evaluated a sedation weaning protocol that included an SBT and found no significant reduction in duration of IMV.[8] However, the main focus of this intervention was the stringent sedative regimen (targeted sedation, arousal assessments, sedation adjustment every 8 hours and sedation weaning). In adults, a Cochrane review of protocolised weaning (17 trials) showed a 26% reduction in duration of IMV in favour of protocols and the most commonly used protocol was daily screening and SBT.[9] Although results from adults cannot be applied directly to the paediatric population, the use of SBT as a weaning strategy shows promise and the paediatric systematic review indicates clinical uncertainty that is worthy of further evaluation.

Various intensive care unit studies have reported associations between rates of high interprofessional collaboration and lower patient mortality[10 11] and improved clinician-to-clinician communication with reductions in length of stay.[12] A team-led approach that improves engagement of all staff in early recognition of readiness and preparation for weaning ventilation has the potential to reduce duration of IMV and PICU length of stay and relieve pressures for beds. As 65% of nurses employed in UK PICU are Band 5 (junior) nurses, this would greatly increase the nursing contribution to the weaning process.[1] Our feasibility study identified very few policies that specifically addressed sedation and weaning guidelines and staff interviews confirmed that a strategy for weaning sedation and ventilation was an important priority in most PICUs.[2] Staff also disclosed continuing uncertainty about readiness to wean, the benefits of an extubation readiness test and its potential impact on duration of IMV in the UK.

The Sedation AND Weaning In CHildren (SANDWICH) trial has the capacity to generate new knowledge on the intervention, its cost-effectiveness and the implementation process. First, it will be large enough to provide reliable evidence for or against a combined ventilator/sedation weaning protocol allowing clear, strong recommendations to be made on the use of this potentially low-cost intervention. Second, it will determine the main organisational and process factors considered important for ensuring the intervention is optimally implemented in PICU.

## METHODS
### Aim and objectives
The SANDWICH trial will evaluate the clinical and cost-effectiveness of a protocol-based intervention incorporating co-ordinated care in managing sedation and weaning ventilation in reducing the duration of IMV in children in PICU. Specific objectives are to determine if the intervention:
► Reduces the duration of IMV in children irrespective of their expected ventilation duration (short or prolonged).
► Reduces length of PICU and hospital stay
► Does not cause additional harm as assessed through review of adverse events and respiratory complications.
► Is cost-effective in the National Health Service.
► Is sustainable and acceptable to staff delivering care.

A process evaluation (PE) conducted alongside the trial will explore the processes involved in delivering the intervention, in order to identify factors and the mechanisms of their interaction that may impact on trial outcomes.

### Study design and setting
#### Setting
SANDWICH is a cluster randomised stepped wedge trial in 18 NHS PICUs. Participating PICUs provide clinical audit data to the Paediatric Intensive Care Audit Network (PICANet) database (www.picanet.org.uk). PICUs will be eligible if they agree to nominate local champions, comply with the protocolised weaning intervention and staff document a willingness to participate in training.

#### Design
The stepped wedge design involves sequential randomised rollout of the intervention over 4-week time periods (see figure 1). Randomisation will be conducted at the hospital site (cluster) level. In general, there is one PICU per site. In one site, there will be two PICUs participating. The site will be treated as one cluster for the purpose of randomisation, and the pair will be randomised to cross from control to intervention together to avoid intervention contamination within the site. In the analysis, we will treat these two PICUs as two separate clusters. This trial requires that all participating PICUs begin the control phase of the trial when the data collection period begins. There will be an initial 8-week period of baseline data

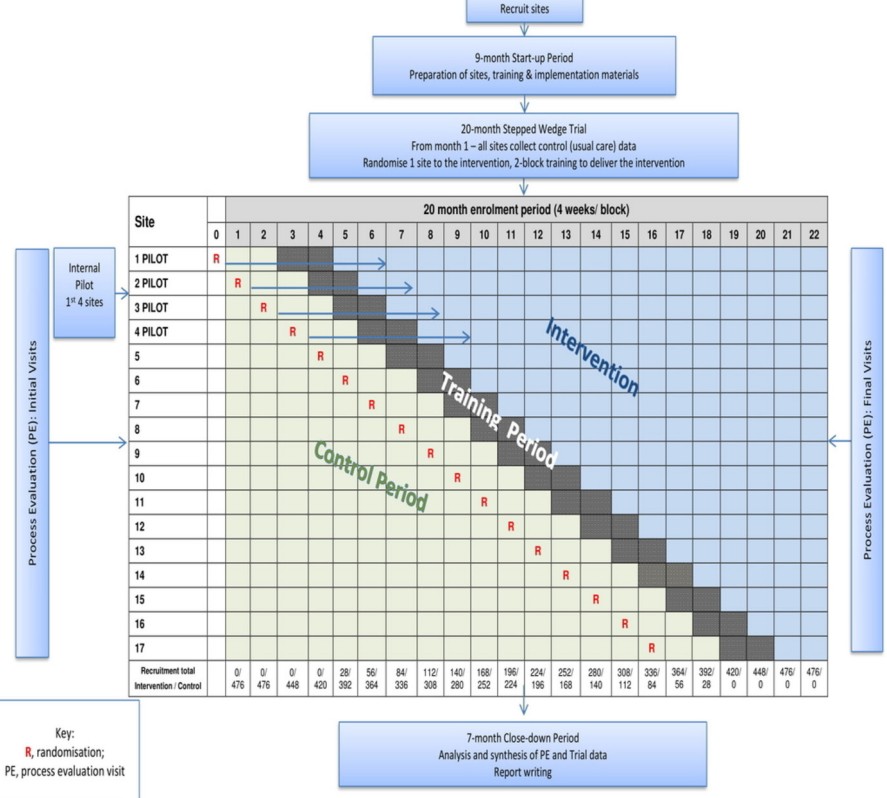

**Figure 1** SANDWICH study flowchart. SANDWICH, Sedation AND Weaning In CHildren.

collection during which the PICU will not be exposed to the intervention. Subsequently, every 4 weeks, one site will be randomly selected to transition to the intervention and start an 8-week training period during which the intervention will be rolled out. The PICU can neither be assumed to be exposed or not exposed during training so in these 8-week periods no patients will be recruited. Once a PICU crosses over to the intervention, it will remain exposed to the intervention for the remaining duration of the study. After the last PICU has crossed over and has fully transitioned to the intervention, there will be a final 8-week period during which all PICUs will be fully exposed.

We have chosen the stepped wedge design over the conventional parallel cluster design for four main reasons. First, we have a limited number of clusters available (maximum 26 PICUs in the UK, but not all likely to agree to participate). With this limited number, the parallel design is infeasible as there are not sufficient clusters to allow detection of the important clinical effect. Second, feasibility work informed us that units are more likely to participate in the trial if they are guaranteed their unit will at some point receive the intervention.[2] Third, it would be infeasible and more costly to deliver the intervention simultaneously to all units randomised to the intervention in a parallel design. Less important factors in our decision process, but none-the-less benefits of this design are the ability to estimate treatment effect heterogeneity (over time and clusters) and it allows for the possibility that the intervention may be tweaked as the trial progresses. This is important as while the intervention

will be clearly documented in accordance with TIDieR guidelines,[13] an intervention that is allowed to adapt to its setting has the best chance of success. Fourth, if the intervention is found to be effective, knowledge translation will be easier as PICUs participating can potentially continue post-trial maximising the benefits of any effects to the NHS and patients.

## Randomisation
The study statistician will conduct the randomisation. Each PICU will be allocated a unique ID. At study commencement, sites will be classified on size based on the number of children receiving IMV in the PICU recorded in the 2017 PICANet database. Randomisation will be balanced on cluster size such that clusters will be randomised in blocks of size 4, with each block containing two large and two small clusters.

## Internal pilot study
An internal pilot in the first four sites randomised to the intervention will evaluate and report progress during the period from randomisation to training, during training and in the 8-week period after implementing the intervention. Specifically, the following criteria will be monitored:
► Actual patient numbers/month of eligible children against predictions.
► Feasibility of data collection procedures.
► Percentage of parents opting out from allowing collection of their child's data.

► Delivery of training (target >80% of staff/unit trained by the end of the pilot period).

► Adherence to intervention components (target >75% by the end of the pilot period).

We will address criteria not achieved in pilot sites through offering support and further training as required. The pilot report will be shared with all sites. The report will inform any actions required in trial management and training to address the above criteria for all sites.

### Timeline

The total study duration will be 36 months to include 9 months for start-up, 20 months for the trial and 7 months for close down.

### Intervention

Sedation and ventilator weaning in standard care will follow current best practice; this is currently non-protocol-based and medically driven. Assessment and management of sedation and ventilator weaning will be according to usual practice. Sedation levels will be assessed and recorded with a validated sedation tool, and ventilator weaning will involve a slow reduction in ventilator support until low levels are achieved consistent with readiness for extubation.

The SANDWICH intervention comprises four components:

► Greater interprofessional collaboration at ward rounds including review of COMFORT scores, sedative regimen and setting targets; and ventilation and setting ventilation goals.

► Sedation measurement using the COMFORT tool.

► Regular daily assessment of criteria for readiness to perform an SBT by bedside nursing staff.

► An SBT and if no distress, a discussion about the decision to extubate.

The intervention training will be delivered at sites by an implementation manager who will train the trainers (local champions, principal investigators and study-specific research nurses). Training will include an online course and face-to-face instruction. A full description of the intervention was available in the study-specific training manual that was only provided to PICUs during and after the training period to avoid influencing practice during the control phase. However, at the time of publication, we are now able to release full details of the intervention, which can be found at http://www.qub.ac.uk/sites/sandwich.

### Patients

All patients admitted to participating PICUs will be screened against the eligibility criteria.

### Inclusion criteria

► All children (<16 years) receiving IMV.

### Exclusion criteria

► Children not expected to reach the primary endpoint (tracheostomy in situ; not expected to survive; treatment withdrawal).

► Children who are pregnant, as documented in their medical notes.

### Consent

A non-confirmed deemed consent (opt-out) approach will be taken in this trial. On patient admission, leaflets will be provided to parents or legal representatives informing them that the PICU is involved in a study and that staff will collect anonymised patient-level information. Leaflets will include contact details for more information or to request that their child's data is not included in the analysis. Individual patient consent will not be confirmed with parents. This deemed consent approach is in line with guidance from the Ottawa Statement,[14] feedback from proposed guidance on consent in cluster trials from the NHS Health Research Authority[15] and was considered appropriate by parents and children during our feasibility work.[16] Posters will be displayed in prominent areas to explain that a trial is taking place in the PICU.

### Patient withdrawal

Children may be withdrawn from data collection on the request of parents or legal representatives. If parents opt-out from the study before data have been collected, this will be noted on the screening log, which will be held at the PICU. Following enrolment, if children are withdrawn, withdrawal will be recorded in the patient record and on PICANet. Data collected up to the point of withdrawal will not be included in the analysis.

### Outcomes

#### Primary outcome

The duration of IMV measured in hours from initiation of IMV (or admission if already intubated) until the first successful extubation (defined as still breathing spontaneously 48 hours following extubation).

#### Secondary outcomes

► Incidence of successful extubation (defined as breathing spontaneously 48 hours following extubation).

► Number of unplanned extubations (defined as dislodgement of the endotracheal tube from the trachea, without the intention to extubate immediately).

► Number of reintubations.

► Total duration of IMV.

► Incidence and duration of postextubation use of non-invasive ventilation.

► Tracheostomy insertion.

► Postextubation stridor.

► Adverse events (eg, unplanned removal/dislodgement of vascular access or non-vascular catheters, bradycardia, hypoxia, cardiopulmonary resuscitation).

- ► PICU length of stay from admission to discharge (in days).
- ► Hospital length of stay from admission to discharge (in days).
- ► Mortality occurring within the ICU.
- ► Mortality occurring within the hospital.
- ► Cost per complication avoided at 28 days

Outcomes will be measured from patient admission up to 90 days or discharge (whichever is earlier). At the end of the 20-month enrolment period, data collection will continue for a maximum of 28 days.

## Data collection

The trial will collaborate with PICANet to make best use of the data collection infrastructure, which exists in PICUs in the UK. Participating PICUs routinely submit clinical data to PICANet to monitor activity and performance. PICUs have full access to and ownership of the data. Data are validated on entry and centrally on the PICANet server. PICANet produce a download facility that allows participating PICUs to extract data required for SANDWICH, thus reducing the burden of data collection for research staff.

When submitting individual patient data to PICANet, research staff will indicate enrolled patients by adding a unique trial number. PICANet has implemented a facility to allow research staff in each PICU to download a pseudoanonymised data set of their data for checking and upload to the SANDWICH Clinical Trials Unit (CTU) as required. This pseudoanonymised data set download will not include patient identifiable information. Trial data will be transmitted from participating PICUs to the CTU electronically using a secure method.

Outcome and compliance data that are not captured by PICANet will be collected and recorded on an electronic case report form (CRF) by PICU research staff and will not include patient identifiable information.

Table 1 shows the patient data collection schedule. The following data are collected:

- Patient characteristics (eligibility, study number, intubation date/time, sociodemographics).
- · Ventilator parameters (mode of IMV, fraction of inspired oxygen, positive end-expiratory pressure (PEEP), peak inspiratory pressure, ventilator rate, tidal volume and the level of pressure support above PEEP).
- · Paediatric Critical Care Minimum Data Set.
- · Adverse events
- ► SANDWICH intervention data (readiness to wean criteria, COMFORT, ward round targets).
- ► Study outcomes.
- ► Post-PICU discharge (hospital length of stay, destination postdischarge, hospital mortality).

## ANALYSIS
### Clinical evaluation

Baseline characteristics will be summarised by exposure and non-exposure to the intervention using summary statistics. PICUs will be classified as being exposed to the intervention on completion of the training period. The primary aim is to evaluate whether there is a difference in the duration of hours on ventilation before and after exposure to the intervention. We will use survival analysis (time to extubation) and estimate an HR for the intervention effect. This means that higher HRs will signify success of the intervention.

We will know exact survival times (ie, times until successful extubation) for most children, but children who die on ventilation, are transferred to another unit, are not weaned before transitioning to the training phase or are not weaned by 90 days will not have a known extubation time. We will treat such events as censored observations, making the assumption that children who are censored for any of these reasons will have an extubation time (ie, were or would have been removed from ventilation) greater than the time until they died or were transferred. These are plausible assumptions. In order to minimise any potential within cluster contamination, we will censor children when their PICU moves into the

| Table 1 | Patient data collection schedule | | | |
|---|---|---|---|---|
| | Baseline (at point of recruitment) | Control phase up to 90 days or PICU discharge | Intervention phase up to 90 days or PICU discharge | Post PICU discharge |
| Patient characteristics | √ | | | |
| Daily 8 am ventilator parameters | | √ | √ | |
| Daily PCCMD | | √ | √ | |
| Daily adverse events | | √ | √ | |
| Outcomes | | √ | √ | |
| 2 hours prior to extubation, ventilator parameters and COMFORT score | | √ | | |
| SANDWICH intervention checklist | | | √ | |
| Hospital discharge and status | | | | √ |

PCCMD, Paediatric Critical Care Minimum Data Set; PICU, paediatric intensive care unit; SANDWICH, Sedation AND Weaning In CHildren.

transition phase. When the PICU moves into the intervention phase, only new admissions will be included.

We will explore various models, but anticipate fitting a Cox proportional hazards model, perhaps with some treatment-by-covariate interaction to incorporate any non-proportionality. Allowance will be made for clustering using a frailty term for each PICU (this is similar to a random effect in a mixed-effects model). We will also adjust for calendar time, since the intervention is sequentially rolled-out. If a child is readmitted or transferred, they will be treated as new events and acknowledged within our analysis. Our primary estimate of the treatment effect will be a cluster and time-adjusted HR along with 95% CIs. Time adjustment is essential because this is a stepped wedge trial.

Secondary analysis will adjust for individual and cluster level covariates (such as the adherence score) and these will be prespecified. Null hypotheses and analyses for secondary outcomes take a similar form to that for the primary outcome. Where outcomes are not survival times, analysis will use the generalised linear mixed model, reporting risk differences for binary outcomes and mean differences for continuous outcomes (all adjusting for cluster and time effects).

Full details of the analyses will be given in the statistical analysis plan.

### Economic evaluation

A cost-effectiveness analysis will be performed from the perspective of the hospital to estimate the cost per complication avoided at 28 days. The occurrence of respiratory complications at 28 days will be measured.

We will estimate total hospital costs until 28 days for each participant by applying appropriate unit costs from the NHS Schedule of Reference Costs[17] to resource use data collected prospectively via the CRF or PICANet, as appropriate. Data on PICU resource use will be obtained via PICANet through the routine collection of the Paediatric Critical Care Minimum Data Set (PCCMDS). The PCCMDS consists of items recorded for each PICU bed-day that can be used to define the level of care and appropriate healthcare resource group. For patients discharged from hospital before 28 days, data on any PICU readmissions within 28 days will come from PICANet but data on readmissions to general hospital wards will not be collected. This is expected to lead to only minimal data loss, as the readmission rate within 30 days in a similar paediatric population was low (5%), with a mean hospital length of stay of less than 1 day.[18]

We will summarise hospital service use, costs and respiratory complications using descriptive statistics. Multilevel mixed-effects regression modelling will be used for total costs and respiratory complications. We will adjust for calendar time and clustering, ensuring consistency with the other models being constructed as part of the main analysis. We will estimate adjusted incremental (differential) total costs and adjusted incremental effects (respiratory complications). Standard methods will be used to explore and display uncertainty in the cost-effectiveness data including scatterplots on the cost-effectiveness plane and cost-effectiveness acceptability curves. Since there is no generally accepted threshold value for cost per respiratory complication avoided, a range of plausible thresholds will be explored. Sensitivity analysis will assess the robustness of the cost-effectiveness results to changes in key parameters. Since the time horizon of the analysis is less than 1 year, it will not be necessary to discount costs and effects.

## PROCESS EVALUATION
### Aim and objectives
The PE will explore the processes involved in delivering the intervention. The specific objectives are:
► To establish the extent to which the intervention is implemented as intended (implementation fidelity), over time and across different PICU.
► To ascertain how PICU staff understand and respond to the intervention, over time and across different PICU.
► To explore the context over time and across different PICU and determine factors (including managerial, economic, organisational and work level) that affect implementation.

### Data collection methods
The methods used for the PE will be:
► Initial site visits to obtain information on context and usual practice collected through interviews and/or focus groups with staff involved in the implementation and delivery of the intervention, using purposive sampling to obtain a range of participants according to grade and profession.
► Telephone interviews with research staff and local champions in the intervention phase to obtain information regarding the implementation process, acceptability of the intervention, barriers and clinical decisions affecting the use of the intervention.
► Final site visits to undertake individual and/or focus group interviews with a purposive sample of staff involved in implementation or intervention delivery. Interviews will explore clinician understanding and experiences, including those relating to barriers and facilitators to the delivery and receipt of the intervention.

### Data analysis methods
Data from the PE will be analysed using the framework approach.[19] A sample of textual data will be reviewed and double-coded by another independent member of the research team to ensure confirmability and trustworthiness. The integration of process and trial outcome data and subsequent analyses will be secondary and explanatory, and separate from the primary effectiveness analysis. The qualitative evidence will be systematically combined with outcome data to identify the processes mediating

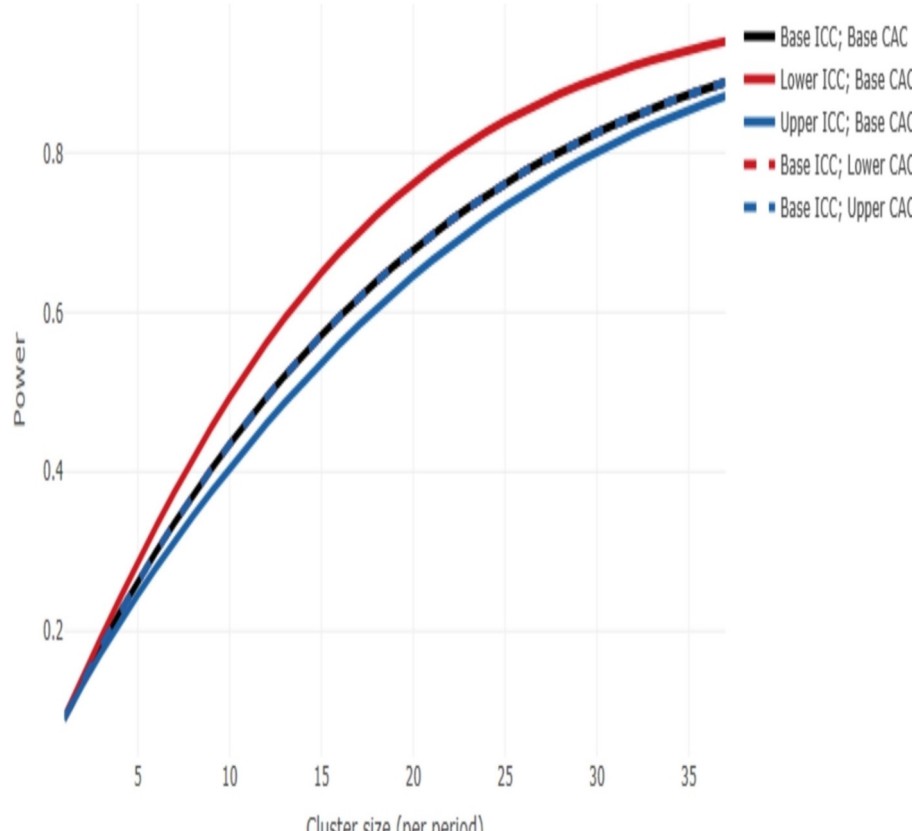

**Figure 2** Power curve. CAC, cluster autocorrelation; ICC, intracluster correlation.

protocol implementation, receipt and setting and observed outcomes.

## SAMPLE SIZE

The primary aim of this study is to determine whether the intervention can reduce the average number of hours on ventilation in eligible children. To inform the power calculation, we used PICU admissions data for the years 2014 to 2016 from units participating in the trial to determine parameters to inform the sample size calculation. The expected sample size is 9520 based on an average cluster size of 28 patients per 4-week block. In this trial, duration of ventilation is censored at the point of transitioning from the control to the training period, discharge to another hospital, at 90 days, death and receiving a tracheostomy, so applying censoring to this data set provided us with a homogeneous population that more accurately reflected the trial population. The mean duration of mechanical ventilation was 5.8 (SD 9.6) days and an Intracluster Correlation (ICC) of 0.005 (95% CI: 0.001 to 0.01). It is postulated that a reduction of 1 day on ventilation is both clinically important and achievable. While our primary analysis will be a survival analysis, no methodology currently exists to determine power in a stepped wedge trial for this outcome type. We therefore determined the power available assuming a continuous outcome. This is expected to be a conservative approach meaning that it should have slightly underestimated the

power not having allowed for the time to event nature of the data.

The cluster sample size app (https://clusterrcts.shin-yapps.io/rshinyapp) was used to update the sample size calculation given this information. Using this app and for the actual design of the trial (using the actual information on the number of clusters and number of steps) and using the following assumptions: no. clusters per sequence=1, ICC=0.005 (with consideration across the range 0.001 to 0.01), an exchangeable correlation structure, mean difference=1, SD=9.6, at 5% significance level, the power is approximately 80% for a cluster size of 28 (figure 2).

## PATIENT AND PUBLIC INVOLVEMENT

We undertook consultation interviews with parents, a 15-year-old PICU survivor and 13 young people who were members of the National Institute for Health Research (NIHR) Clinical Research Network: Children, Young Person's Advisory Group about the proposed trial. Their views have contributed to the choice of patient relevant outcomes and informed the approach to consent. The consultation work was funded by the Northern Ireland Health and Social Care Research and Development Division and aided by Jenny Preston, Consumer Liaison Manager for the NIHR-Children Research Network. We secured patient and young people's continued involvement to provide advice on study design, implementation, parent and child

information leaflets, assistance with preparation of educational materials and dissemination of findings. Father and son, Lewis and Archie Veale (now 18 years), agreed to be on the Trial Steering Group for this study. They have first-hand experience of the difficulties of ventilator weaning (Archie spent 8 weeks in PICU in 2014).

## ETHICS, OVERSIGHT AND DISSEMINATION
### Oversight
The Northern Ireland CTU (NICTU) will manage the trial. The Trial Management Group, chaired by the chief investigator, will meet monthly and have responsibility for the day-to-day operational management of the trial. The Trial Steering Committee (TSC) will meet approximately every 6 to 12 months and provide oversight for the conduct of the study on behalf of the Funder (National Institute for Health Research) and Sponsor (Queen's University Belfast). The Data Monitoring Committee (DMC) will meet approximately every 6 to 12 months and will safeguard the rights, safety and well-being of trial participants; monitor data and make recommendations to the TSC on whether there are any safety reasons why the trial should not continue; and monitor overall study conduct to ensure validity and integrity of the study findings.

### Dissemination
We will publish findings from this study in a timely and relevant manner to influence health service policy to deliver public benefit. Our dissemination strategy targets a variety of service users including: (1) the UK paediatric intensive care community (trial updates at the PICS Study Group meetings); (2) the wider paediatric intensive care community (presentations at national and international meetings; publications in high-quality peer-reviewed open access journals); (3) the public via a final report in the NIHR Health Technology Assessment (HTA) journal and national parent support and liaison groups, via social media and through the PICS families group; and (4) NHS managers and commissioners if the study supports a change of practice.

## TRIAL STATUS
This paper presents the protocol (version 5, 12 March 2019). The trial began on 5 February 2018. At the time of first manuscript submission, data collection for the trial was ongoing and due to be complete in October 2019. The trial results will be disseminated in 2020 through presentations at national and international conferences and publication in peer-reviewed medical journals.

## DATA STATEMENT
The data generated and/or analysed during the SANDWICH trial are not yet publicly available due to the ongoing nature of the trial. When the trial is complete, data sets will be available from the chief investigator on reasonable request and arrangements will be made to deposit them in a suitable online repository.

**Author affiliations**
[1] Wellcome-Wolfson Institute for Experimental Medicine, School of Medicine, Dentistry and Biomedical Sciences, Queen's University Belfast, Belfast, UK
[2] Northern Ireland Clinical Trials Unit, Belfast, UK
[3] Centre for Public Health, Institute of Clinical Sciences, Queen's University Belfast, Belfast, UK
[4] Public Health, Epidemiology and Biostatistics, Institute of Applied Health Research, College of Medical and Dental Sciences, University of Birmingham, Birmingham, UK
[5] Paediatric Intensive Care Unit, Royal Brompton Hospital, London, UK
[6] Paediatric Intensive Care Unit, Birmingham Women's and Children's Hospital, Birmingham, UK
[7] Faculty of Medicine and Health, University of Leeds, Leeds, UK
[8] Paediatric Intensive Care Unit, Great Ormond Street Hospital for Children NHS Trust, London, UK
[9] Institute of Child Health, University College London, London, UK
[10] Faculty of Health and Applied Sciences, University of the West of England Bristol, Bristol, UK
[11] MRC Centre for Inflammation Research, The Queen's Medical Research Institute, The University of Edinburgh, Edinburgh, UK

**Acknowledgements** We thank the following people for their contributions to the set-up and delivery of the SANDWICH trial: Lynn Murphy, NICTU Manager, Pauline Bradley, Data Manager, Gerard O'Hanlon Data Manager and Ruth Holman, Clinical Trial Administrator; and Dr Katherine Fielding and Professor Gavin Perkins for agreeing to chair the TSC and DMC, respectively. We thank the Paediatric Intensive Care Society—Study Group for their ongoing advice and support of this trial. We also thank the research and clinical staff from the 17 participating sites: 1. Alder Hey Children's Hospital, Liverpool. 2. Royal Belfast Hospital for Sick Children, Belfast. 3. Birmingham Children's Hospital, Birmingham. 4. Bristol Royal Children's Hospital, Bristol. 5. Royal Brompton Hospital, London. 6. Addenbrooke's Hospital, Cambridge. 7. Noah's Ark Children's Hospital for Wales, Cardiff. 8. Great Ormond Street Hospital, London. 9. Variety Children's Hospital, King's College London. 10. Leeds General Infirmary, Leeds. 11. Royal Victoria Infirmary, Newcastle. 12. John Radcliffe Hospital, Oxford. 13. Southampton General Hospital, Southampton. 14. St George's Hospital, London. 15. St Mary's Hospital, London. 16. Royal Stoke University Hospital, Stoke-on-Trent. 17. Sheffield Children's Hospital, Sheffield.

**Contributors** BB, AA, MC, KH, JJ, DM, DFMA, CMD, KM, MM, RB, RP, MJP, LNT and TW conceived the SANDWICH trial. BB led the grant application and, as Chief Investigator, has oversight for the trial. KH and CMD have oversight for the statistical analysis; AA has oversight for the economic analysis; LMI and TW designed the online training materials; JJ designed and conducted the process evaluation; and RB and MM managed the trial. BB drafted this manuscript, and all authors contributed to, read and approved the final version.

**Funding** This project was funded by the National Institute for Health Research (NIHR) Health Technology Assessment (HTA) Programme (project number: HTA—15/104/01). Queen's University Belfast is the sponsor for the trial.

**Disclaimer** The views and opinions expressed therein are those of the authors and do not necessarily reflect those of the HTA Programme, NIHR, NHS, the Department of Health nor the sponsor.

**Competing interests** None declared.

**Patient consent for publication** Not required.

**Ethics approval** The protocol (and amendments) received ethical approval from NRES Committee East Midlands—Nottingham 1 REC (17/EM/0301).

**Provenance and peer review** Not commissioned; externally peer reviewed.

**ORCID iDs**
Bronagh Blackwood http://orcid.org/0000-0002-4583-5381
Ashley Agus http://orcid.org/0000-0001-9839-6282

Mike Clarke http://orcid.org/0000-0002-2926-7257
Daniel Francis McAuley http://orcid.org/0000-0002-3283-1947

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
