## [Reviewer comments · BMJ Open]

ARTICLE DETAILS

TITLE (PROVISIONAL)	Sedation AND Weaning In CHildren (SANDWICH): protocol for a cluster randomised stepped wedge trial.
AUTHORS	Blackwood, Bronagh; Agus, Ashley; Boyle, Roisin; Clarke, Mike; Hemming, Karla; Jordan, Joanne; Macrae, Duncan; McAuley, Daniel; McDowell, Clíona; McIlmurray, Lisa; Morris, Kevin; Murray, Margaret; Parslow, Roger; Peters, Mark; Tume, Lyvonne N.; Walsh, Tim

VERSION 1 - REVIEW

REVIEWER	M-Sánchez Hospital General Universitario de Elche (España) Enfermera doctora por la Universidad de Alicante
REVIEW RETURNED	29-May-2019

GENERAL COMMENTS	REFERENCE: In the article there are a total of 19 bibliographical references, of which 10 have an antiquity of more than 5 years.
--

REVIEWER	Felizitas A Eichner Institute for Clinical Epidemiology and Biometry University of Würzburg Germany
REVIEW RETURNED	14-Jul-2019

GENERAL COMMENTS	This protocol is well-written, presents clear objectives and well thought out methods. Please find a few minor remarks below. Abstract p.2 line 45 In the abstract "18 [participating] paediatric care units" are mentioned, but in Figure 1, only 17 PICUs are shown. The Design section states that two PICUs at the same site will be randomised together. If this were the case in the study, it would be good to make it clear in the figure, which PICUs were randomized together. Otherwise, the numbers appear to be contradicting each other. Ethics and dissemination: p2. line 58 The Ethics committee approval is referenced as 17/EM/0301, on page 11, line 26 as 7/EM/0301. Which one is correct?
---

	Methods Adverse events: One of the secondary outcomes is adverse events. Please give some more detail at least once in the paper which events will be counted as adverse events in this trial. Stepped-wedge cluster randomized trial design: It would be interesting for the reader to get a short insight into why the authors used the stepped-wedge cluster randomized trial design instead of a simple cluster-randomized design, e.g. certain ethical or methodological reasons. Internal pilot phase: p. 5 line 48 and following A pilot phase is a very useful tool to, for example, test study procedures and to get a valid estimate of eligible patients and the recruitment rate. However, I was wondering how the results of the internal pilot phase were implemented into the main trial of SANDWICH. Maybe the authors can give a little more detail on this. Consent: p. 6 line 44 and following The authors state that the deemed consent approach is in line with the Ottawa statement, [...] and the NIHS Health Research Authority, but more importantly it should be stated explicitly if this opt-out approach was also approved by the REC. Anonymised vs. pseudonymised data p. 6 line 48 and following The authors state towards the parents that anonymised patient-level information is collected, but it is possible to identify a patient if parents opt-out so that this data can be excluded from the analysis. It is of course very important that patients remain identifiable during the study, but the statements are somewhat contradictory. It would be great if authors could clarify this further: Is it in fact a fully pseudonymised data collection or will data only be collected in a pseudonymised manner and then be anonymised before the final analysis? Intervention: referring to review checklist no. 4 This is only intended as a remark. Presumably, the authors only wanted to state little detail on the intervention as the trial is still running. It would however be very interesting to read more about the specifics of the intervention in the main paper, e.g. details of the SBT and the methods used to increase greater inter-professional collaboration, to promote transparency and reproducibility.
--	--

REVIEWER	Katherine Harding La Trobe University / Eastern Health Australia
REVIEW RETURNED	15-Jul-2019

GENERAL COMMENTS	This is a well written trial protocol for a cluster randomized stepped wedge trial. I preface this review by saying that the experience that
--

	informs this review is in the methodology rather than the clinical content of the trial. The authors appear to provide sound justification for their use of outcome measures, but my capacity to comment on the clinical detail of choices regarding this aspect of the trial is limited. All aspects of the trial are clearly described. There were just a few minor issues that I felt could be further clarified: Design, page 5: "In sites where two PICUs will be participating, the pair will be randomized to cross from control to intervention together" Does this mean that the data from these pairs will be combined and treated as a single PICU? Or are they treated in the analysis as separate PICUs? If they are treated as two separate PICUs, how does this impact on the randomization process? At how many sites is this likely to be an issue? Please clarify. In Figure 1 there is no indication that any two sites commence together. Internal pilot study, Page 5: Will any action be taken based on outcomes of the internal pilot study? What happens, for example, if delivery of training targets are not met during this period? Intervention, page 6: It would be helpful in this section to include a couple of sentences on what is happening during the pre-intervention period. Eg "During the pre-intervention period all sites will provide usual care.....this is expected to vary from site to site but typically includes....." Analysis, page 8: It was not entirely clear to me how the issue of censoring participants is being handled during the transition phase. While I agree that it is important to manage the issue of participants who potentially cross from one period to the other, the process and justification need to be clearly explained. Are participants being recruited right up until the final day of the pre intervention period? Given the total duration of IMV is the primary outcome this would appear to have the potential to impact on the findings of the trial, particularly given that the sample size has been calculate using a mean duration of IMV of 6 days, with a one day change deemed to be meaningful. Do you also plan to censor patients still on IMV at the end of the post intervention period, so that this issue impacts on both pre and post intervention participants equally?
--	---

REVIEWER	GONZALO HERNANDEZ MARTINEZ HOSPITAL VIRGEN DE LA SALUD, TOLEDO, SPAIN
REVIEW RETURNED	30-Jul-2019

GENERAL COMMENTS	Interesting protocol aiming to reduce time under invasive mechanical ventilation implementing a 4 components intervention in an ongoing cluster-randomized study. The protocol is correct to me, and results will probably change clinical practice. I have some questions:  - Inclusion criteria: all children under 16 receiving IMV. Will patients under IMV for an expected short time under IMV (Eg. surgical intervention) be included in the analysis? Sedation and weaning is clearly different in this subgroup of patients.
--

	- The SANDWICH intervention training is not totally explained. Can authors detail the training? Any leader? - I am not sure a greater inter-professional collaboration is necessary to review clinical aspects like sedative regimen, setting ventilation goals... Many training staf in the PICUs recruiting? SANDWICH seems to me like a first approach to weaning. Results will probably be positive.
--	--

REVIEWER	SIMONE PIVA University of Brescia
REVIEW RETURNED	26-Aug-2019

GENERAL COMMENTS	The presented protocol is well written; objectives are well stated. There are no major comments on my side.
---

VERSION 1 – AUTHOR RESPONSE

Reviewer(s) Reports:

Reviewer: 1

Reviewer Name: M-Sánchez

Institution and Country: Hospital General Universitario de Elche (España) Enfermera doctora por la Universidad de Alicante:

Please state any competing interests or state 'None declared': None declared

Please leave your comments for the authors below

REFERENCE:

In the article there are a total of 19 bibliographical references, of which 10 have an antiquity of more than 5 years.

We have cited 5 references older than 10-years that we felt were important to include for the following reasons: one is a trial that reached inclusion criteria for the Cochrane systematic review (Maloney) and therefore had to be cited. The other four are important seminal papers that address the association between inter-professional teamwork (as opposed to individual discipline group) and patient outcomes (Wheelan); association between coordinated care in ICU and patient outcomes (Knaus); a coordinated clinician communication protocol to reduce CBSI (Pronovost); and a conceptual framework for implementation fidelity that has been used in other studies of process evaluations (Carroll). We do not feel it is necessary to add additional references.

Reviewer: 2

Reviewer Name: Felizitas A Eichner

Institution and Country: Institute for Clinical Epidemiology and Biometry University of Würzburg Germany Please state any competing interests or state 'None declared': None declared

This protocol is well-written, presents clear objectives and well thought out methods. Please find a few minor remarks below.

Abstract

p.2 line 45

In the abstract “18 [participating] paediatric care units” are mentioned, but in Figure 1, only 17 PICUs are shown. The Design section states that two PICUs at the same site will be randomised together. If this were the case in the study, it would be good to make it clear in the figure, which PICUs were randomized together. Otherwise, the numbers appear to be contradicting each other.

Response: Thank you for pointing this out. We have amended the text on the flowchart from PICU to Site. Because this is the protocol paper, we cannot identify the randomised site that had two PICUs in the flowchart because the randomisation process was undertaken each month and we did not know in advance when this site would have been randomised. However, we will keep this comment in mind for the results paper. To clarify the discrepancy between the 18 sites and the 17 sites for the reader, we have inserted the following into the text under study design and setting on p.5:

“In one site there will be two PICUs participating. The site will be treated as one cluster for the purpose of randomisation and the pair will be randomised to cross from control to intervention together to avoid intervention contamination within the site. In the analysis we will treat these two PICUs as two separate clusters.”

Ethics and dissemination:

p2. line 58

The Ethics committee approval is referenced as 17/EM/0301, on page 11, line 26 as 7/EM/0301. Which one is correct?

Response: Apologies for the typo the correct number is 17/EM/0301 and we have amended this in the ethics section in the manuscript.

Adverse events:

One of the secondary outcomes is adverse events. Please give some more detail at least once in the paper which events will be counted as adverse events in this trial.

Response: We have included examples in the secondary outcomes section on p.7 which reads as follows:

- Adverse events (e.g. unplanned removal/dislodgement of vascular access or non-vascular catheters; bradycardia; hypoxia; cardiopulmonary resuscitation)

Stepped-wedge cluster randomized trial design: It would be interesting for the reader to get a short insight into why the authors used the stepped-wedge cluster randomized trial design instead of a simple cluster-randomized design, e.g. certain ethical or methodological reasons.

Response: We have included a paragraph on our reasons taken from our grant application and inserted it into the section study design and setting as follows. This necessitated adding an additional reference [13] and the reference list was amended.

“We have chosen the stepped wedge design over the conventional parallel cluster design for four main reasons. First, we have a limited number of clusters available (max. 26 PICUs in the UK, but not all likely to agree to participate). With this limited number the parallel design is infeasible as there are not sufficient clusters to allow detection of the important clinical effect. Second, feasibility work informed us that units are more likely to participate in the trial if they are guaranteed their unit will at some point receive the intervention.[2] Third, it would be infeasible and more costly to deliver the intervention simultaneously to all units randomised to the intervention in a parallel design. Less important factors in our decision process, but none-the-less benefits of this design are the ability to estimate treatment effect heterogeneity (over time and clusters) and it allows for the possibility that the intervention may be tweaked as the trial progresses. This is important as whilst the intervention will be clearly documented in accordance with TIDieR guidelines [13], an intervention that is allowed to adapt to its setting has the best chance of success. Fourth, if the intervention is found to be effective, knowledge translation will be easier as PICUs participating can potentially continue post trial maximising the benefits of any effects to the NHS and patients.”

Internal pilot phase:

p. 5 line 48 and following

A pilot phase is a very useful tool to, for example, test study procedures and to get a valid estimate of eligible patients and the recruitment rate. However, I was wondering how the results of the internal pilot phase were implemented into the main trial of SANDWICH. Maybe the authors can give a little more detail on this.

Response: As this is the protocol, we cannot include information on how the results of the internal pilot phase were implemented into the main trial of SANDWICH. However, we have added a sentence to explain the action we planned if criteria were not met. This is inserted in the internal pilot study section as follows:

“We will address criteria not achieved in pilot sites through offering support and further training as required. The pilot report will be shared with all sites. The report will inform any actions required in trial management and training to address the above criteria for all sites.”

Consent:

p. 6 line 44 and following

The authors state that the deemed consent approach is in line with the Ottawa statement, [...] and the NIHS Health Research Authority, but more importantly it should be stated explicitly if this opt-out approach was also approved by the REC.

Response: This section refers to consent procedures only; the REC approvals are detailed in the ethics section on p.11

Anonymised vs. pseudonymised data p. 6 line 48 and following

The authors state towards the parents that anonymised patient-level information is collected, but it is possible to identify a patient if parents opt-out so that this data can be excluded from the analysis. It is of course very important that patients remain identifiable during the study, but the statements are somewhat contradictory. It would be great if authors could clarify this further: Is it in fact a fully pseudonymised data collection or will data only be collected in a pseudonymised manner and then be anonymised before the final analysis?

Response: We are unsure of your question because we feel we have addressed this clearly in the data collection section. Sites normally enter patient data into the PICANet database (of course this cannot be pseudoanonymised at this point). For the purpose of this study, PICANet implemented a facility for sites to download a pseudoanonymised dataset of their data for checking and upload to the SANDWICH Clinical Trials Unit. This pseudoanonymised dataset does not include any patient identifiable information (e.g. date of birth, name, address). We hope that our insert clarifies this point on p.8:

“This pseudoanonymised dataset download will not include patient identifiable information.”

Intervention:

referring to review checklist no. 4

This is only intended as a remark. Presumably, the authors only wanted to state little detail on the intervention as the trial is still running. It would however be very interesting to read more about the specifics of the intervention in the main paper, e.g. details of the SBT and the methods used to increase greater inter-professional collaboration, to promote transparency and reproducibility.

Response: We could not provide details earlier in the protocol, however as the last site has crossed over to the intervention we have released the details on the SANDWICH website. We included a sentence to indicate this under the intervention section as follows:

“A full description of the intervention was available in the study-specific training manual that was only provided to PICUs during and after the training period to avoid influencing practice during the control phase. However, at time of publication, we are now able to release full details of the intervention which can be found at <http://www.qub.ac.uk/sites/sandwich>”

Reviewer: 3

Reviewer Name: Katherine Harding

Institution and Country: La Trobe University / Eastern Health Australia Please state any competing interests or state 'None declared': None declared

This is a well written trial protocol for a cluster randomized stepped wedge trial. I preface this review by saying that the experience that informs this review is in the methodology rather than the clinical content of the trial. The authors appear to provide sound justification for their use of outcome measures, but my capacity to comment on the clinical detail of choices regarding this aspect of the trial is limited.

Response: Thank you for your complements on our design.

All aspects of the trial are clearly described. There were just a few minor issues that I felt could be further clarified:

Design, page 5: "In sites where two PICUs will be participating, the pair will be randomized to cross from control to intervention together" Does this mean that the data from these pairs will be combined and treated as a single PICU? Or are they treated in the analysis as separate PICUs? If they are treated as two separate PICUs, how does this impact on the randomization process? At how many sites is this likely to be an issue? Please clarify. In Figure 1 there is no indication that any two sites commence together.

Response: Only one participating site had two PICUs. As outlined in our protocol, this site was treated as one cluster for the purpose of randomisation. However, in the analysis we will treat these two units as two separate clusters. This will reflect the higher correlation within each of these separate units (as opposed to the combination of the two of them). We have changed the text on page 5 to make this clearer:

"In one site there will be two PICUs participating. The site will be treated as one cluster for the purpose of randomisation and the pair will be randomised to cross from control to intervention together to avoid intervention contamination within the site. In the analysis we will treat these two PICUs as two separate clusters."

We have amended the text on the flowchart from PICU to Site. Because this is the protocol paper, we cannot identify in the flowchart the site that had two PICUs because the randomisation process was undertaken each month and we did not know in advance when this site would have been randomised. However, we will keep this comment in mind for the results paper.

Internal pilot study, Page 5: Will any action be taken based on outcomes of the internal pilot study? What happens, for example, if delivery of training targets are not met during this period?

Response: We would have addressed this with individual site and offered support or further training where needed. It's not possible in the protocol to foresee every possibility, but we have added a sentence to explain the action we planned if criteria were not met. This is inserted in the internal pilot study section as follows:

“We will address criteria not achieved in pilot sites through offering support and further training as required. The pilot report will be shared with all sites. The report will inform any actions required in trial management and training to address the above criteria for all sites.”

Intervention, page 6: It would be helpful in this section to include a couple of sentences on what is happening during the pre-intervention period. Eg "During the pre-intervention period all sites will provide usual care.....this is expected to vary from site to site but typically includes....."

Response: Yes of course, we have now included some detail on usual practice in this section as follows:

“Sedation and ventilator weaning in standard care will follow current best practice; this is currently non-protocol-based and medically-driven. Assessment and management of sedation and ventilator weaning will be according to usual practice. Sedation levels will be assessed and recorded with a validated sedation tool and ventilator weaning will involve a slow reduction in ventilator support until low levels are achieved consistent with readiness for extubation.”

Analysis, page 8: It was not entirely clear to me how the issue of censoring participants is being handled during the transition phase. While I agree that it is important to manage the issue of participants who potentially cross from one period to the other, the process and justification need to be clearly explained. Are participants being recruited right up until the final day of the pre intervention period? Given the total duration of IMV is the primary outcome this would appear to have the potential to impact on the findings of the trial, particularly given that the sample size has been calculate using a mean duration of IMV of 6 days, with a one-day change deemed to be meaningful. Do you also plan to censor patients still on IMV at the end of the post intervention period, so that this issue impacts on both pre and post intervention participants equally?

Response: Yes, we will censor patients still on IMV at the end of the intervention period, so that this issue impacts on end of control period and end of intervention period participants equally.

Reviewer: 4

Reviewer Name: GONZALO HERNANDEZ MARTINEZ Institution and Country: HOSPITAL VIRGEN DE LA SALUD, TOLEDO, SPAIN Please state any competing interests or state 'None declared': NO CONFLICTS OF INTEREST

Interesting protocol aiming to reduce time under invasive mechanical ventilation implementing a 4 components intervention in an ongoing cluster-randomized study.

The protocol is correct to me, and results will probably change clinical practice. I have some questions:

- Inclusion criteria: all children under 16 receiving IMV. Will patients under IMV for an expected short time under IMV (Eg. surgical intervention) be included in the analysis? Sedation and weaning is clearly different in this subgroup of patients.

Response: Yes, all patients will be included. We agree with you that short term MV patients probably don't require a weaning process. In fact, the commissioning call for this study asked for a study of patients expected to require longer term ventilation. We gave this serious consideration prior to designing the protocol because to introduce a time limit on short or long term was not possible –

particularly as the primary outcome is duration of ventilation and the intervention is applied from day 1. For example, if we defined long term as 'expected to be ventilated >24 hours' and the intervention worked, then the patient would move from the 'long term' time period to the 'short term' time period classification and we would not have a robust trial. To circumvent this, we have pre-specified a list of conditions and diagnostic codes that typically require a short duration of ventilation (identified using historic PICANet data). We will undertake our analysis on all patients and additionally, using this list, we will undertake a sensitivity analysis of the data without the patients in the short term classification group. To do this at the end of the trial was an easier option to take, because to ask research staff to screen and classify patients might create much work and might result in problems in inconsistency.

, but

- The SANDWICH intervention training is not totally explained. Can authors detail the training? Any leader?

Response: We could not provide details in the protocol, however as the last site has now crossed over to the intervention we have released the details on the SANDWICH website. We included a sentence to indicate this under the intervention section as follows:

"A full description of the intervention was available in the study-specific training manual that was only provided to PICUs during and after the training period to avoid influencing practice during the control phase. However, at time of publication, we are now able to release full details of the intervention which can be found at <http://www.qub.ac.uk/sites/sandwich>"

- I am not sure a greater inter-professional collaboration is necessary to review clinical aspects like sedative regimen, setting ventilation goals... Many training staff in the PICUs recruiting? SANDWICH seems to me like a first approach to weaning. Results will probably be positive.

Response: The sedative regimen and setting ventilation goals does not necessarily need collaborative review, but these important aspects need to be discussed and communicated if both doctors and nurses are to work towards the same goal of getting the patient in the right ready for a spontaneous breathing trial. In our feasibility work, we found that there was a distinct lack of communication, and this, coupled with the literature in this area, convinced us to include it as a component in the intervention. The process evaluation findings will shed light on whether or not staff find this component a crucial or useful part of the intervention.

Reviewer: 5

Reviewer Name: SIMONE PIVA

Institution and Country: University of Brescia Please state any competing interests or state 'None declared': None

The presented protocol is well written; objectives are well stated. There are no major comments on my side.

Response: Many thanks for this very positive views.